# Experimental Biomechanics of Neonatal Brachial Plexus Avulsion Injuries Using a Piglet Model

**DOI:** 10.3390/bioengineering12010091

**Published:** 2025-01-20

**Authors:** Anita Singh, Kalyani Ghuge, Yashvy Patni, Sriram Balasubramanian

**Affiliations:** 1Bioengineering Department, Temple University, Philadelphia, PA 19122, USA; kalyani.prabhakar.ghuge@temple.edu; 2North Creek High School, Bothell, WA 98012, USA; 1077559@apps.nsd.org; 3School of Biomedical Engineering, Science and Health Systems, Drexel University, Philadelphia, PA 19104, USA; sb939@drexel.edu

**Keywords:** avulsion, neonatal, brachial plexus, biomechanics, strain, load, stretch, palsy, birthing

## Abstract

Background: A brachial plexus avulsion occurs when the nerve root separates from the spinal cord during birthing trauma, such as shoulder dystocia or a difficult vaginal delivery. A complete paralysis of the affected levels occurs post-brachial plexus avulsion. Despite being reported in 10–20% of brachial plexus birthing injuries, it remains poorly diagnosed during the acute stages of injury, leading to poor intervention approaches. The poor diagnosis of brachial plexus avulsion injury can be attributed to the currently unavailable biomechanics of brachial plexus avulsion. While the biomechanical properties of neonatal brachial plexus are available, the forces required to avulse a neonatal brachial plexus remain unknown. Methods: This study aims to provide detailed biomechanics of the required forces and corresponding strains for neonatal brachial plexus avulsion. Biomechanical tensile testing was performed on an isolated, clinically relevant piglet spinal cord and brachial plexus complex, and the required avulsion forces and strains were measured. Results: The reported failure forces and corresponding strains were 3.9 ± 1.6 N at a 27.9 ± 6.5% strain, respectively. Conclusion: The obtained data are required to understand the avulsion injury biomechanics and provide the necessary experimental data for computational model development that serves as an ideal surrogate for understanding complicated birthing injuries in newborns.

## 1. Introduction

Spinal nerve root avulsions are reported in 2% to 5% of brachial plexus injuries (BPIs) [1]. The reported spinal nerve root avulsions can cause concomitant spinal cord injury (SCI) that is often misdiagnosed in clinical settings [2,3,4,5]. The inaccurate or delayed diagnosis of concomitant SCI can be attributed to the lack of understanding of the biomechanics of root avulsion injuries, thereby leading to limited options for acute treatment.

Brachial plexus (BP) originates from the ventral rami of C5 through T1 spinal nerves and is a complex network of nerves transmitting motor and sensory signals to the upper extremities, including the shoulder, arm, and hand [6,7,8,9]. Brachial plexus injuries, resulting from accidents/trauma in adults to complicated birthing in newborns, qualify as one of the most debilitating injuries affecting the upper extremities [10]. Types of reported BPIs include overstretched BP, pre-ganglionic or post-ganglionic BP ruptures, BP root avulsion injuries, or combined injuries. BP avulsion occurs when the roots of a BP nerve are completely separated from the spinal cord due to overstretching or traction forces applied on the BP in response to overextension of the neck [11]. While the events leading to BP injuries, such as the separation of the head and the shoulder of the subject during the upper lesion (Erb’s palsy) and upper limb forcefully abducting above the head during the lower lesion (Klumpe’s Palsy), are well described, the biomechanical responses of BP during these injuries remains poorly understood [12]. Furthermore, there is limited literature available on the failure responses during BP rupture and avulsion injuries; they are reported only in adult human cadaveric and animal studies. The biomechanics of avulsion injuries in neonates remain unknown [13]. Additional studies are needed to provide insight into the biomechanics of neonatal BP root avulsion injuries since the forces responsible for BP avulsion injuries might also be responsible for concomitant SCI and are critical not only for developing injury prevention strategies but also for proper injury management in neonates.

The current literature on the tensile biomechanical response of the human BP tissue is limited to adult human cadaveric tissues [14,15,16,17,18,19]. Furthermore, most animal studies have also reported failure load data in small adult animal models, including rats and rabbits [20,21,22,23,24,25]. Although the available studies provide a framework for understanding the BP’s mechanical failure response when subjected to stretch, the reported values are primarily reported in adult human and small animal models. No studies have been performed in neonates. The available studies have reported that neonatal nerves are immature, with thinner axons and less myelination [26,27]. These attributes might contribute to differences in the biomechanical properties of peripheral nerves in neonates.

In neonates, brachial plexus avulsion occurs when the nerve root separates from the spinal cord during birthing trauma, including an obstructed vaginal delivery such as shoulder dystocia [28,29]. A complete paralysis of the affected levels occurs post-BP avulsion. Despite being reported in 10–20% of brachial plexus birthing injuries, such injuries remain poorly diagnosed during the acute stages, leading to poor intervention approaches [30]. Furthermore, since concomitant SCI can also occur during BP avulsion injuries, understanding the biomechanics of neonatal brachial plexus avulsion injuries is critical for not only proper prognosis but also for optimal treatment strategy. The ethical limitations associated with conducting experiments using human neonatal tissue warrant studies in neonatal animal models, preferably a large animal model, to serve as promising surrogates that can help understand the biomechanics of neonatal BP avulsion injuries. This study aims to fill this critical research gap by providing detailed biomechanics of the required forces and corresponding strains for neonatal brachial plexus avulsion injuries using a large neonatal piglet animal model.

## 2. Materials and Methods

In this in vitro study, a total of six spinal cord segments (C3-T2) with intact bilateral brachial plexus complexes were obtained from six normal neonatal piglets (3–5 days old) immediately post-partum. A summary of the study flow is shown in Figure 1.

### 2.1. Tissue Harvesting

The Institutional Animal Care and Use Committee (IACUC) approved all the surgical procedures used in this study. All euthanasia procedures were conducted humanely and in accordance with the AVMA Panel on Euthanasia guidelines and per the approved protocol using an overdose of euthasol (0.4 mL/kg). Immediately post-euthanasia, the spinal cord at C3–T3 levels was exposed using the posterior approach. Briefly, a midline incision was made over the cervical and upper thoracic spine. The muscles and soft tissue were retracted to expose the spinous processes at the C3–T2 cervical and thoracic spinal levels. The lamina was then removed, and the spinal cord was exposed after removing the ligamentum flavum and the underlying epidural fat. The animal was then placed in a supine position to expose the brachial plexus bilaterally. With the upper limbs in abduction, the axillary region was exposed by making a midline incision through the skin and fascia overlying the trachea down to the upper third of the sternum. The superior and inferior flaps were released using blunt dissection, and the cervical and thoracic segments of the entire BP complex were exposed. The intact BP complexes were then isolated from the surrounding muscles and connective tissues and carefully examined to locate the bifurcations of the BP division segment (M shape). BP segments were then identified relative to these bifurcations. The segments closer to the spine were identified as the root/trunk, and those below these bifurcations were identified as the cord/nerve segments. The spinal cord and the bilaterally attached BP segments until the terminal nerve branches were carefully harvested and preserved in phosphate-buffered saline until testing, which was performed within two hours after tissue harvest.

### 2.2. Test Apparatus

Biomechanical avulsion testing was performed using a custom-built mechanical testing setup that consisted of a linear actuator (to subject stretch on BP segment), a load cell (to measure the mechanical load sustained during avulsion injury), a clamp (to secure the BP tissue to the actuator), and a 3D imaging system (to measure in situ strain on the BP segment during stretch), as shown in Figure 2 [31].

An 8 × 8-inch aluminum square anchored to a cart served as the base support for the apparatus. A two-foot-tall pole was threaded into the aluminum base with a connecting unit attached at the top. This connecting unit could swivel and was linked to the linear-moving actuator, enabling both angular and linear measurements of the actuator’s movement. The apparatus was designed to allow free adjustment to accommodate varying sample sizes. A load cell was integrated between the linear actuator and clamps to record force data [21].

The stereo imaging system employed a ZED Mini camera positioned above the mechanical setup. This passive stereo camera comprises two horizontally aligned lenses separated by 63 mm. The 3D points of the sample displacement were obtained from both the left and right lenses, and the direct linear transformation method was used to calculate nerve displacement data [32].

### 2.3. Biomechanical Testing

Two custom-built clamps were used to secure the isolated spinal cord at both ends while nerve root avulsion testing was performed. The three identified upper, middle, and lower BP trunks (upper, middle, and lower) were then prepared for testing by isolation and clamping. The clamped BP segment, including the roots and the trunk, and the adjacent spinal cord tissue were then marked with Indian ink while utilizing a grid to ensure consistency among the various tests, and the ZED Mini stereo camera was positioned above them to capture images of these markers during the stretch (Figure 3). These images were later analyzed to determine the in situ tissue strain during stretch.

Data recording and triggering the mechanical setup and camera were managed using customized MATLAB code. The actuator stretched the BP root/trunk segments at a rate of 500 mm/min until complete failure occurred based on previously reported studies [31]. Load and displacement data were recorded at 1000 Hz, while images were captured at 100 frames per second (FPS). After the test, the clamps were loosened to inspect whether the tissue had experienced avulsion or if any nerve slippage from the clamps had occurred. Only data obtained from successfully avulsed samples were analyzed.

### 2.4. Data Analysis

Strain analysis was performed using the DLTdv Digitizing tool, and the displacements of the markers in two dimensions were obtained from the left and right lenses of the ZED Mini stereo camera. Using the DLT calibration coefficients, 3D points were calculated from the image tracking of the 2D image points. Then, an open source MATLAB code, DLTcal5.m [33], was used to calibrate the ZED Mini using the DLT calibration method [32]. Then, a MATLAB (Version 9.7) code was used to import the dataset and calculate the length of the BP segment (*l*), defined as the section between the insertion and the clamp. The distance ($l_{i}$) between each adjacent marker at each time point (*i*) during the tensile testing was calculated using Equation (1).(1)$$l_{i}=\sqrt{{\left(x_{2i}-x_{1i}\right)}^{2}+{\left(y_{2i}-y_{1i}\right)}^{2}+{\left(z_{2i}-z_{1i}\right)}^{2}\;}$$

In this equation, *l_i_* is the distance between two markers at any time point (*i*) during the tensile testing. *x*_1_, *y*_1_, and *z*_1_ are the 3D set of points marking distance at the previous time point (Figure 3). *x*_2_, *y*_2_, and *z*_2_ are the 3D set of points marking the distance at the time point considered. The change in lengths between time points ($∆l_{i}$) can be calculated using Equation (2).(2)$$∆l_{i}=l_{i}-l_{0}$$

Here, *l_i_* is the distance at the chosen time interval, and *l*_0_ is the distance at the initial time point. Using the result from this equation, the percent strain of the clamped nerve was determined (Equation (3)). This characterized the percentage change in length at any two of the time intervals chosen.(3)$$percent\;strain\;=\;\frac{∆l_{i}}{l_{0}}\times 100$$

The displacement, load, image, and time data were recorded synchronously and used to plot the load vs. time and strain vs. time plots. The maximum load and the corresponding strain were identified from these plots. The changes in structural integrity and failure location (proximal, mid-length, distal) of the tested tissue were determined using the images obtained from the camera. Non-avulsion testing was excluded from this study.

### 2.5. Statistical Analysis

Statistical analysis was performed using SPSS software (Version 29.0.2.0, Chicago, IL, USA). Values for maximum load and corresponding strains for each of the three tested BP trunk levels (upper, middle, and lower) were expressed as mean ± standard deviation (mean ± Stdev). The normality of the data was assessed before applying the appropriate statistical tests. Based on the observed normality, the data were compared using a one-way ANOVA followed by pairwise comparisons conducted using independent *t*-tests. A *p* value less than 0.05 was considered significant.

## 3. Results

Out of 36 tested samples, 2 samples slipped during tensile testing and were excluded from the data analysis. In the remaining 34 samples, 80% reported avulsion injuries, and others were ruptured at the clamp. A total of eight avulsion injuries were reported in the upper trunk, ten in the middle trunk, and nine in the lower trunk BP segments.

The reported average and standard deviation values for the maximum load and corresponding strain are summarized in Table 1.

The avulsion strain and loads reported from the biomechanical tensile testing of the piglet spinal cord and BP trunk complex are shown in Figure 4 and Figure 5. Strain analysis reported avulsion strains of 31.1 ± 5.73% for the upper trunk (n = 8), 25.2 ± 5.43% for the middle trunk (n = 10), and 28.0 ± 8.54% for the lower trunk (n = 9) BP segments. No significant differences in the avulsion strains were reported between the three tested levels.

The avulsion loads reported in the upper, middle, and lower trunks of the BP segments were 2.4 ± 0.56 N (n = 8), 4.1 ± 1.1 N (10), and 5.4 ± 1.9 N (n = 9), respectively. Significantly higher avulsion loads were reported in the lower trunk BP segments when compared to the upper trunk BP segments. No other differences were found in the avulsion loads between the other tested BP trunk segments. The observed higher avulsion loads in the BP upper trunk segment can be attributed to the anatomical characteristics of the BP complex.

## 4. Discussion

The accurate diagnosis of traumatic BP avulsion injury is often delayed due to a poor understanding of the biomechanics of such injuries. Although the available adult human cadaveric and small animal studies provide a framework for understanding the BP’s mechanical failure responses when subjected to stretch, the reported values for BP avulsion injuries in neonates remain unknown. Such data are critical to developing preventative and interventional strategies.

The available human and animal studies provide a wide range of average failure force and average elongation in an intact BP complex when stretched. Human studies have reported the failure load and strain of BP when stretched at 200 mm/min to be 630 N (range: 365–807 N) and 37% (range: 23–53.5%), respectively [16]. Since BP injuries can be rupture, avulsion, or combination, studies have also reported factors that affect the injury type (i.e., avulsion or rupture). One such reported factor was the loading direction that affected the type of injury in an intact BP complex. When stretching intact BP complexes perpendicular to the midline of the spine, the loading force resulted in a weakening of the connections between the epineurium and transverse processes, making the intact BP complex more vulnerable to avulsion-type injuries (88%) at the nerve root levels [16,34]. In the current study, the trunk was clamped and stretched perpendicular to the midline of the spine. Avulsion injuries were reported in 80% (27/34) of cases. While the percentages of the observed injury type between the previously reported and current studies are similar, with 88% and 80% of avulsion-type injuries, respectively, the reported failure load in the current study is several folds lower, with a reported avulsion load of 3.9 ± 1.6 N at 27.9 ± 6.9% strain. The lower avulsion load and strain values can be attributed to species-specific and age-specific differences. Previous studies have reported higher ultimate load and stress in human cadaveric tissue when compared to adult animal models, with failure loads being 16.9 ± 2.7 N and strain 24.0 ± 1.1% in an adult rabbit animal model [35]. Also, age-dependent differences have been confirmed, such that the ultimate stress of an adult (age range: 20–69 years) sciatic is 1.28 ± 0.016 kg/mm^2^ compared to adolescents (age range: 0–19 years) at 1.14 ± 0.035 kg/mm^2^ and in neonates (age range: one month) at 0.96 ± 0.026 kg/mm^2^ [36].

In an animal study, Takai et al. (2002) studied the lower BP trunk of an adult rabbit animal model. The trunk was stretched to failure at a rate of 10 mm/min [35]. The reported average values for maximum tensile force, ultimate tensile stress, ultimate strain, and elastic modulus were 16.9 ± 2.7 N, 6.9 ± 0.39 MPa, 24.0 ± 1.1%, and 28.5 ± 1.8 MPa, respectively [35]. The reported values for avulsion loads and strains of the lower trunk in the current study are 5.4 ± 1.9 N and 28.0 ± 8.5%, respectively. While similar failure strain values are reported, the failure load values are much lower in the neonatal animal model when compared to the adult animal model. These findings align with the previously reported age-specific differences.

The current study is the first to report avulsion injury forces and strains in a neonatal animal model. Such information is critical for understanding the injury mechanisms that contribute to the effective management of the incidence of neonatal avulsion injuries. The ethical limitations associated with conducting experiments using human neonatal tissue warrant the development of novel methodologies to investigate neonatal BP tensile biomechanical properties. Studies using neonatal large animal models serve as promising surrogates. Singh et al. (2018), using a neonatal piglet animal model (3–5 days old), reported the biomechanical properties of various BP segments [21]. This study further utilizes the approach to report avulsion forces and strain in the neonatal BP–spinal cord complex. Data obtained from these studies not only provide a comprehensive understanding of the neonatal BP and associated spinal cord responses to tensile stretch but are also very critical to the development of existing computational models that serve as a promising surrogate for understanding complicated birthing scenarios that lead to brachial plexus injuries in neonates [37,38]. Future studies can focus on loading directions and underlying molecular mechanisms of avulsion injuries. The findings from these studies can directly impact clinical practice by guiding the development of prevention and treatment strategies for neonatal delivery-related injuries.

## Figures and Tables

**Figure 1 bioengineering-12-00091-f001:**
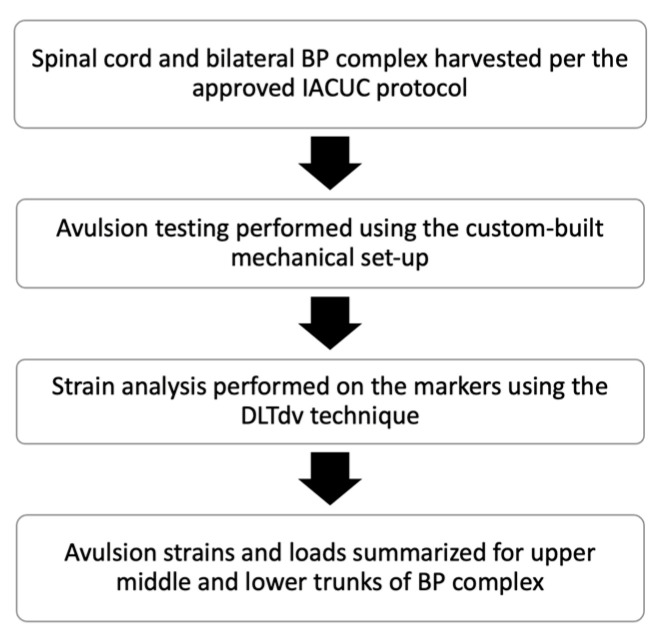
Study flowchart detailing a summary of the steps involved in this study.

**Figure 2 bioengineering-12-00091-f002:**
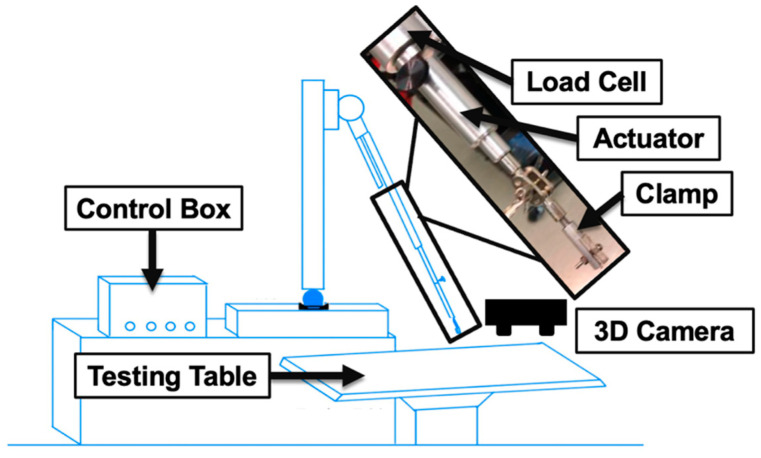
Biomechanical testing setup details. The custom-built setup includes a control box, load cell, actuator, and clamps. A 3D camera system placed above the testing sample acquired the images for the strain analysis.

**Figure 3 bioengineering-12-00091-f003:**
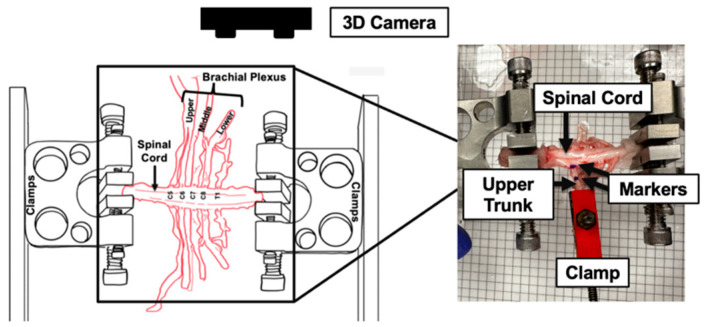
Biomechanical testing of the spinal cord–BP complex. Ink markers placed on the spinal cord adjacent to the rootlets and root/trunk segment of the BP were tracked for strain analysis.

**Figure 4 bioengineering-12-00091-f004:**
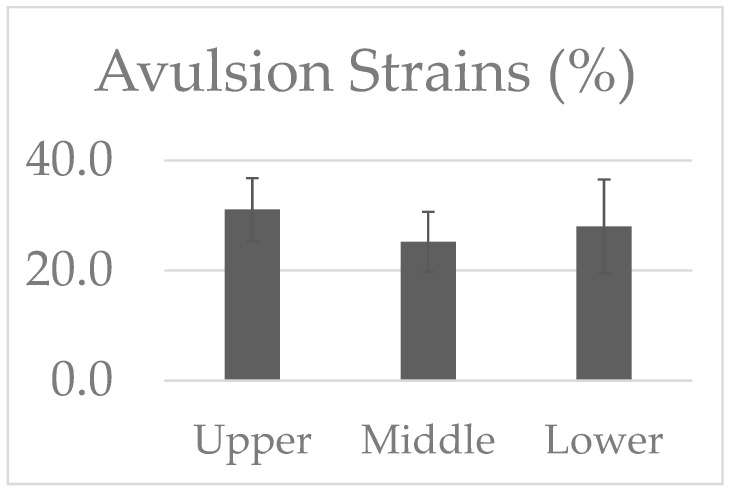
Avulsion strains (%) reported in the tested upper, middle, and lower trunks of the BP segments.

**Figure 5 bioengineering-12-00091-f005:**
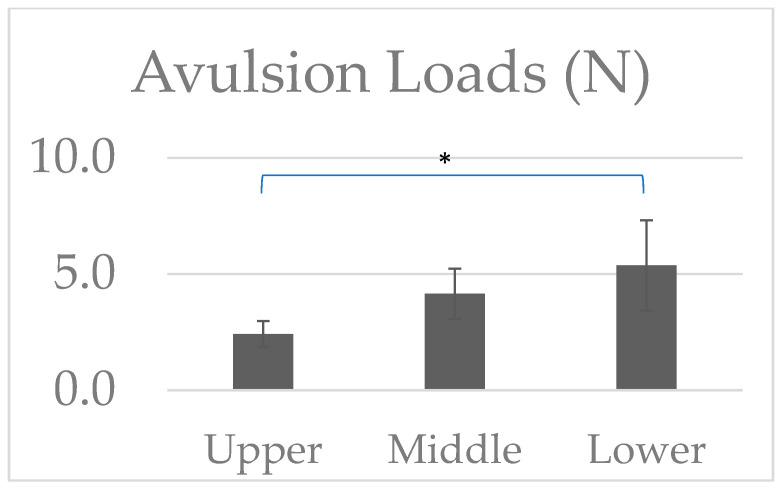
Avulsion loads (N) reported in the tested upper, middle, and lower trunks of the BP segments. *: *p* < 0.05 indicates a significant difference between the groups.

**Table 1 bioengineering-12-00091-t001:** Avulsion strains and loads values (average ± standard deviation (Stdev)) reported from in vitro tensile testing of spinal cord–BP complex.

Total # of Samples	Avulsion Strain (%)		Avulsion Load (N)	
	Average	Stdev	Average	Stdev
27	27.9	6.5	3.9	1.6

## Data Availability

Due to privacy/ethical restrictions, the data can only be made available upon request.

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
