# Peer review of "Experimental Biomechanics of Neonatal Brachial Plexus Avulsion Injuries Using a Piglet Model"

_bioengineering, 2025, doi:10.3390/bioengineering12010091_

Round 1
Reviewer 1 Report
Comments and Suggestions for Authors
Dear all,
Thank you for the opportunity to review this manuscript. The manuscript aligns with the goal of the Bioengineering, and the topic offers information for researchers, professionals, and the Biomedical Engineering and Biomaterials section. The manuscript titled presents an important topic in biomechanics and neonatal injury prevention. I believe the information provided might need to be clarified. Consequently, some points are listed below:
Title
I suggest: ‘Experimental Biomechanics of Neonatal Brachial Plexus Avulsion Injuries Using a Piglet Model’.
1. Introduction
I wish if the authors could state why neonatal tissues are biomechanically distinct (e.g., structural immaturity).
Try to avoid redundancy, the authors might need to combine paragraphs 3 and 4 which declares the neonatal biomechanics differ from adults and animal models.
2. Materials and Methods
I suggest utilizing a flowchart of the study, which will streamline the readability of the research process.
The sample size appears to be missing. This section should be included.
Line 136; ‘at a rate of 500 mm/min until complete failure occurred’, please provide confirmation for this rate ‘500 mm/min’, was this based on prior studies or preliminary testing? The sentence needs a reference to support its ideas.
It would be more benefit to specify the MATLAB version.
2.4. Data Analysis; please provide details on the calibration process, such as the calibration grid or reference object used to compute the DLT coefficients, as the validation of the calibration process will further strengthen the analysis.
The used equations, it is standard and accurate method for determining marker displacements in 3D space, however, please clarify, how could you ensure that the markers were placed consistently across samples, as marker placement errors can affect measurements.
Equations 1, 2, and 3: the equations are presented clearly, but variables need to be defined after their introduction (e.g., li, is the distance between any two markers at any time point. x1i, y1i, z1i are …….).
Lines 165-169; in the statistical analysis, please clarify whether you assessed normality of data distribution before applying ANOVA.
Results
In Figures 3 and 4: Label the markers of significantly different between the groups (e.g., * indicates p < 0.05), it appears in Fig 4, but without indicating the P value.
Discussion
The discussion highlights the significance of the findings, but the authors need to discuss with the data of other studies similar to the recent study and models.
References
References and Literature in a research paper should not be older than 10 years old (Ref2 1997, Ref3 1998, Ref8 1990, and Ref18 1993); it should supplement the existing body of literature with some recent sources.
The reference publication date should bold and after the journal name, please use the ACS style guide to be compatible with Bioengineering’s journal guidelines.
Best wishes,
Comments on the Quality of English Language
NA
Author Response
Title
Comment 1: I suggest: ‘Experimental Biomechanics of Neonatal Brachial Plexus Avulsion Injuries Using a Piglet Model’.
Response 1: We thank the reviewer for the recommendation and have incorporated the new proposed title in our revised manuscript.
Introduction
Comment 2: I wish if the authors could state why neonatal tissues are biomechanically distinct (e.g., structural immaturity).
Response 2: We thank the reviewer for this comment and have included studies that reported neonatal nerve to be immature with thinner axons and less myelination (Lines 60-62).
Comment 3: Try to avoid redundancy, the authors might need to combine paragraphs 3 and 4 which declares the neonatal biomechanics differ from adults and animal models.
Response 3: We agree with the reviewer and have revised the two paragraphs to limit redundancy and for a better read.
Materials and Methods
Comment 4: I suggest utilizing a flowchart of the study, which will streamline the readability of the research process.
Response 4: We thank the reviewer for this suggestion and have included a flowchart of the study (New Figure 1).
Comment 5: The sample size appears to be missing. This section should be included.
Response 5: We thank the reviewer for this suggestion and have included the missing sample size information (Line 181, Lines 190-191, and Line 194).
Comment 6: Line 136; ‘at a rate of 500 mm/min until complete failure occurred’, please provide confirmation for this rate ‘500 mm/min’, was this based on prior studies or preliminary testing? The sentence needs a reference to support its ideas.
Response 6: We have included the requested change (Lines 142-143).
Comment 7: It would be more benefit to specify the MATLAB version.
Response 7: We have included the details of the MATLAB version in the revised manuscript (Line 153).
Comment 8: 2.4. Data Analysis; please provide details on the calibration process, such as the calibration grid or reference object used to compute the DLT coefficients, as the validation of the calibration process will further strengthen the analysis.
Response 8: We thank the reviewer for this suggestion and have included the details on calibration steps in the paper (Lines 150-152).
Comment 9: The used equations, it is standard and accurate method for determining marker displacements in 3D space, however, please clarify, how could you ensure that the markers were placed consistently across samples, as marker placement errors can affect measurements.
Response 9: We agree with the reviewer’s comments on consistency in marker placement for strain analysis. We utilized a grid for marker placement across each test (visible in the background of new Figure 3). We have added this detail in the revised manuscript (Lines 133-134).
Comment 10: Equations 1, 2, and 3: the equations are presented clearly, but variables need to be defined after their introduction (e.g., li, is the distance between any two markers at any time point. x1i, y1i, z1i are …….).
Response 10: We thank the reviewer for this comment and made the revisions as suggested (Lines 155, 157).
Comment 11: Lines 165-169; in the statistical analysis, please clarify whether you assessed normality of data distribution before applying ANOVA.
Response 11: We thank the reviewer for this clarifying comment. We assessed the normality of data to ensure we could perform appropriate statistical tests. Details of this approach are now added in the required section (Lines 175-179).
Results
Comment 12: In Figures 3 and 4: Label the markers of significantly different between the groups (e.g., * indicates p < 0.05), it appears in Fig 4, but without indicating the P value.
Response 12: We thank the reviewer for this comment. No significant difference was observed in the avulsion strains data (old Fig 3 and new Fig 4). For old Fig 4 (new Fig 5) avulsion loads, we have included the p-value details per the reviewer’s suggestion (Line 205).
Discussion
Comment 13: The discussion highlights the significance of the findings, but the authors need to discuss with the data of other studies similar to the recent study and models.
Response 13: We thank the reviewer for this comment, but the thorough literature review performed by the authors indicated very limited literature on avulsion injury strains and loads. Furthermore, such data is missing in the neonatal population. We have compared the findings with those reported in adult studies.
References
Comment 14: References and Literature in a research paper should not be older than 10 years old (Ref2 1997, Ref3 1998, Ref8 1990, and Ref18 1993); it should supplement the existing body of literature with some recent sources.
Response 14: We thank the reviewer for this comment and have removed older references.
Comment 15: The reference publication date should bold and after the journal name, please use the ACS style guide to be compatible with Bioengineering’s journal guidelines.
Response 15: The referencing style has now been corrected.
Reviewer 2 Report
Comments and Suggestions for Authors
The authors present a study entitled “Biomechanics of Neonatal Brachial Plexus Avulsion Injuries”. A fixed stretching speed (500 mm/min) was used in the experiment, but there may be many different load conditions and directions in actual delivery, which does not fully simulate the actual situation. However, this study still can give some information to understand the biomechanics of neonatal brachial plexus avulsion injuries. After reviewing this manuscript, I have some comments listed as follows:
1) Lines 224~226, 2 in the unit should be superscripted (kg/mm2).
2) Some text is cut off in Figure 4. Please correct it.
3) Many studies use animal models to investigate neonatal brachial plexus palsy, and most of them adopt rats or rabbits as the model. The authors used newborn pigs as experimental subjects, which may be more appropriate. Nevertheless, the authors should cite and discuss some literature related to neonates.
4) Although the conclusions note the potential uses of the data, the authors do not explicitly recommend further research or further emphasize how this research could directly impact clinical practice, like prevention and treatment of neonatal delivery-related injuries.
Author Response
Comment 1: Lines 224~226, 2 in the unit should be superscripted (kg/mm2).
Response 1: We thank the reviewer for this comment and have made the recommended changes (new Lines 233-235).
Comment 2: Some text is cut off in Figure 4. Please correct it.
Response 2: We thank the reviewer for this comment and have made the recommended changes (new Fig. 5).
Comment 3: Many studies use animal models to investigate neonatal brachial plexus palsy, and most of them adopt rats or rabbits as the model. The authors used newborn pigs as experimental subjects, which may be more appropriate. Nevertheless, the authors should cite and discuss some literature related to neonates.
Response 3: We thank the reviewer for this suggestion. Our Pubmed search from 1977 to 2025 resulted in only one article that reported the biomechanical studies on neonates (Singh et al.), and no neonatal biomechanical study was available on concomitant spinal cord avulsion injuries during BP injury. Our discussion includes the currently available literature.
Comment 4: Although the conclusions note the potential uses of the data, the authors do not explicitly recommend further research or further emphasize how this research could directly impact clinical practice, like prevention and treatment of neonatal delivery-related injuries.
Response 4: We thank the reviewer for this comment and have now included the future scope of the study and its findings in developing strategies to help prevent and treat delivery-related neonatal injuries.